# Experimental Evaluation of LoRa in Transit Vehicle Tracking Service Based on Intelligent Transportation Systems and IoT

**Felipe Jurado Murillo** [1][iD]**, Juan Sebastián Quintero Yoshioka** [1][iD]**, Andrés David Varela López** [1][iD]**, Ricardo Salazar-Cabrera** [2,*][iD]**, Álvaro Pachón de la Cruz** [1][iD] **and Juan Manuel Madrid Molina** [1][iD]

1   Information Technology and Telecommunications Research Group (I2T), ICT Department, Universidad Icesi, Cali 760001, Colombia; felipe.jurado@u.icesi.edu.co (F.J.M.); juan.quintero12@u.icesi.edu.co (J.S.Q.Y.); andres.varela@u.icesi.edu.co (A.D.V.L.); alvaro@icesi.edu.co (Á.P.d.l.C.); jmadrid@icesi.edu.co (J.M.M.M.)
2   Telematics Engineering Research Group (GIT), Telematics Department, Universidad del Cauca, Popayán 190001, Colombia
*   Correspondence: ricardosalazarc@unicauca.edu.co; Tel.: +57-313-586-0304

**Abstract:** Long-range (LoRa) technology is a low power wide area network (LPWAN) technology that is currently being used for development of Internet of things (IoT)-based solutions. Transit transport, mainly in medium-sized cities where transit vehicles do not have exclusive lanes, is a service that can be improved with a tracking service using technology such as LoRa. Although some proposals exist, there is not enough experimental information to validate the LoRa technology as adequate. This article: (a) evaluates the operation of LoRa technology in a transit vehicle tracking service in a medium-sized city, based on an Intelligent Transportation Systems architecture and IoT; and (b) investigates optimal LoRa technology configuration parameters for the service. Experiments were performed in a semi-controlled environment using LoRa devices and a gateway, by measuring the received packets and the receive signal strength indicator (RSSI) and modifying: (a) distance; (b) number of devices; and (c) the main LoRa transmission parameters. Obtained results show the ideal values of parameters vary considerably with distance and number of devices used. There were very few settings of the experiments in which the RSSI and packet levels were adequate while distance and number of devices were both changed.

**Keywords:** LoRa; Intelligent Transportation System; IoT; transit vehicle tracking service; experimental evaluation

## 1. Introduction

Low power wide area network (LPWAN) technologies are being used recently for development of IoT-based solutions due to good range, efficiency, low energy consumption and low operation costs [1–7]. Long-range (LoRa), narrowband IoT (NBIoT) [8], Sigfox [9], and other LPWAN technologies have been proposed for a wide variety of services in which the connection of many devices over long distances is required [10].

Transit transport, mainly in medium-sized cities, where transit vehicles do not have exclusive lanes, has drawbacks related to speeding and noncompliance with traffic regulations, vehicle routes, schedules and frequencies [11,12]. These issues can be minimized by controlling and tracking transit vehicles through an IoT-based solution, using technologies such as LPWAN to transmit information between the vehicles and the information centers.

Among the LPWAN technologies, LoRa has been selected as suitable for mobility services (e.g., transit vehicle tracking service) due to its high range, low operation cost, use of unlicensed frequency spectrum, adequate data rate and low latency [3,4].

LoRa is adequate for communication between vehicles and information centers in a transit vehicle tracking service, in terms of frequency of messages, size of each message, and latency. A tracking service does not require a high frequency of messages, a maximum of 15 messages per minute from each vehicle is enough for locating a vehicle with enough accuracy; the principal data in each message are coordinates (latitude and longitude) that account for a relatively short message (which could also be encoded to reduce its size) [2]; and the latency of each of the messages is not critical, since the location of a vehicle can be reported with up to 1 s of delay without a considerable effect. LoRa technology has a high enough data rate to handle the required message rate; the size of the packet, although limited, is adequate for this type of service; and latency is low enough to have transmission delays of less than 1 s in the distances handled in a tracking service [3,4].

Implementing LoRa technology in a large-scale transit vehicle tracking system in an intermediate city in a developing country such as Colombia (or in intermediate cities in other countries, with similar transportation systems) involves a high risk, no evidence was found in industry or academy on the implementation of this technology for this type of services in a city. Although there are some related works [2,6,13,14], there is no practical evidence regarding the number of end devices that can be served by a single gateway, maintaining the quality of service. In addition, there is no detailed analysis of the adequate values for the LoRa transmission parameters (spreading factor, bandwidth and coding rate), to achieve the best transmission efficiency at certain distances and with a certain number of LoRa devices.

Additionally, no architecture has been adopted for design and development of most of these mobility services. An architecture is necessary for an adequate standardization and integration with other types of mobility services or smart cities services. Intelligent transportation systems (ITS) are a viable option to achieve an adequate standardization between the mobility services of a city; having an ITS architecture as a basis for the design of the tracking service is a relevant factor for guaranteeing a major improvement of mobility in a city [15,16].

In addition to an adequate ITS architecture for, the service was considered as an IoT-based solution, because the transit vehicle tracking service requires a device in each vehicle, which will sense a number of variables and transmit them to a central server. Information collected by these devices allows to perform an adequate control of transit vehicles and other related services, e.g., traveler information and traffic control.

LoRa's communication performance can be tuned by modifying several physical layer settings, including transmission power (TP), carrier frequency (CF), bandwidth (BW), spreading factor (SF), and coding rate (CR) [17,18]. In a field test environment, where the equipment (LoRa devices and gateways) is already selected, it is difficult to modify the TP and CF parameters in the tests, because the devices are already factory-configured to use a fixed set of TP and CF values; however, it is more feasible to modify SF, BW and CR in each test. SF is the ratio between the symbol and chip rates. SF can take integer values ranging from 6 to 12. BW is the width of frequencies in the transmission band. A typical LoRa network operates at a BW of either 500 kHz, 250 kHz or 125 kHz. Finally, CR is the forward error correction (FEC) rate used by the LoRa modem that offers protection against bursts of interference, and can be set to either 4/5, 4/6, 4/7 or 4/8 [18].

In the surveyed related work [2,10,13,17–33], although some experiments are performed using LoRa (in some cases using only LoRa transmission technology, and in some cases using the LoRa Wide Area Network protocol, LoRaWAN), most of these works are not focused on the application of this technology in mobility services. Furthermore, in works such as [2,10,13,19–33] a wide range of possible settings (i.e., specific sets of SF, BW and CR values) are not explored. In works such as [17,18,23], although numerous of possible settings is evaluated, the characteristics of the performed experiments, such as distance between devices, number of devices, packet rate, or stationary devices,

means that these works do not have a similar context to this work, which is focused on a transit vehicle tracking service.

This research evaluated the operation of LoRa in a transit vehicle tracking service in an intermediate city (based on an ITS architecture and IoT), and investigated the optimal LoRa technology configuration parameters for the service. Research was conducted in four stages: (a) determining an adequate ITS architecture of the tracking service; (b) designing and developing a prototype of the service; (c) designing and developing experiments to evaluate LoRa in the prototype; (d) obtaining the optimal LoRa parameter values for the service.

In the authors' previous works [13,15], stage (a) was developed in detail, for this reason the results from [13,15] are used in this work. Regarding stage (b), a proof of concept was developed in [13], so this work makes an incremental contribution to the development of the system, moving from proof of concept to a prototype. In [13], five very basic experiments using LoRa were designed and developed, only two devices were used and only one of the setting parameters (spreading factor) was varied within a limited set of values; in contrast, this work includes much more robust experiments for finding the optimal values of the LoRa configuration parameters. The number of devices was increased (up to four devices in some experiments) and all the 72 possible settings obtained by combining all possible values of the three LoRa parameters were evaluated. Therefore, the greatest contribution of this work is made in stages (c) and (d).

Considering the limitations of the studies of LoRa technology reviewed in the literature (limited number of parameter combinations and lack of sufficient field tests), experiments were performed in a semi-controlled environment using LoRa devices and a gateway in the 915 MHz radio band (band allocated to LoRa in the American continent). In these experiments, the number of received packets at the gateway and the receive signal strength indicator (RSSI) were measured while modifying the transmission distance, the number of devices used, and the main LoRa transmission parameters. SF was varied from 7 to 12; BW was set to either 125 kHz, 250 kHz, or 500 kHz; CR values of 4/5, 4/6, 4/7 and 4/8 were used.

The technical novelty of our research can be identified in the following aspects: (a) the characteristics of the experiments performed to evaluate LoRa are different from those executed in the related works; (b) the prototype proposed for the monitoring system is based on an ITS architecture that was designed considering internationally recognized references; and (c) the analysis of the results of the experiments, with the help of comparati ve graphs and tables, to facilitate reading and understanding for the user.

The remainder of this paper is organized as follows: Section 2 presents the related works; Section 3 presents the materials and methods used in the project, including the description of the performed LoRa evaluation experiments; Section 4 presents the experimental results; Section 5 discusses the results, and Section 6 presents the conclusions of the study.

## 2. Related Works

In the literature review, many works related to the use of LoRa were found in a wide variety of services that involve the use of wireless technologies.

In works such as [10,13,17–27,29–32] the operation of LoRa technology in a variety of services is evaluated, using basic messages at the link layer, without any support at the network layer. In contrast, works such as [2,14,19,28,33] evaluated the operation of LoRa using the LoRaWAN network protocol. The LoRaWAN network protocol increases communication security and allows operational adaptability of some parameters, however it introduces some limitations regarding the use of certain frequencies (because it must be used on specific channels) and packet size [14].

Regarding the type of services evaluated with LoRa technology, some reviewed works [2,13,14,21,24,33] are related to tracking transportation services (the type of service evaluated in this paper), however many of them [10,17–20,22,23,25–32] are related to other services or a specific service is not mentioned.

In works such as [2,10,13,17–33], the authors perform some experiments to evaluate LoRa performance, however in very few cases a sufficient number of tests is performed, by varying parameters such as distance, number of devices and LoRa configuration settings.

In [18], Bor et al. use a large number of possible settings of LoRa, with different spreading factors, bandwidth settings, coding rates and transmission powers. The authors performed a detailed experimental analysis of LoRa transmission parameter settings on energy consumption and communication reliability. In the experiments they developed, they used a sender and a receiver located in an office building, at a distance of 50 m and with some walls in between. The sender transmitted 255 packets on each transmission setting. The experiment cycles used 1152 settings, SF (from 7 to 12), BW (125 kHz, 250 kHz and 500 kHz), CR (4/5, 4/6, 4/7 and 4/8) and TP (2 dBm to 17 dBm). Although numerous settings were used in this work (many more than those used in our work), the characteristics of the experiments (in terms of number of devices, packet rate, and distance between device and gateway) are different in our context, also the work was focused on energy consumption.

In [23], the authors also use a considerable number of settings, but use a simulator to test the different settings to examine and understand scalability of LoRa networks. Although a great variety of settings are used in the tests of this evaluated work, when using a simulator (instead of field tests), the results obtained could be ruling out some important aspects that affect the operation of LoRa. Additionally, the characteristics of the experiments do not conform to those of a transit vehicle tracking service.

In [17] the authors used 18 settings, varying the values of the LoRa technology parameters (SF, CR, BW and bit rate). The tests used three types of scenarios: indoor, outdoor, and underground, with distances between a few meters and 165 m and a packet rate of 1 every 5 s. Although the test environment tries to emulate a timely report of a typical IoT sensor for urban monitoring, the performed experiments vary considerably in aspects such as the total number of settings, the distances and the displacement of the devices, with respect to the present paper.

Works related to mobility services [2,13,14,21,24,33] performed field tests, mostly using vehicles traveling on certain types of routes, but have important limitations in terms of the number of evaluated settings. Additionally, some works use a limited number of devices, and in some others speed variation is not considered as a factor that can affect the operation of LoRa.

The authors in [24] state that LoRa can be effectively utilized for moderately dense networks of very low traffic devices not having strict latency or reliability requirements. Because of this, LoRa can be used for a transit vehicle tracking service. However, Ref. [24] features no field tests.

In [25], the authors focus mainly on the relationship between range and scalability of LoRa. They state a single gateway can cover a range of a few kilometers in an urban area and up to 30 km in a suburban area (considering typical densities of end devices). Ref. [25] considers the Doppler effect for moving devices, with a propagation factor of 12; they conclude that at a relative speed greater than 40 km/h, performance impairment in communication is evident. For speeds of up to 25 km/h communication is reliable enough, which allows its use in people monitoring and tracking applications. Thus, the experiments performed in the present paper take into account a 25 km/h speed limit to avoid degradation due to Doppler effect.

In the literature revision, no work was found that integrated the characteristics of the present research. These integrated characteristics are what determine the novelty of this research. The work presented in this paper proposes a prototype of a transit vehicle tracking service, based on ITS and using LoRa technology in the critical communication that occurs between vehicles and network devices located on the roads. The proposed technology is evaluated through six experiments that use a considerable number of settings, some of these experiments (the last three) emulate the environment the evaluated service may have (in terms of packet rate, routes traveled, vehicle speeds and number of devices operating simultaneously).

## 3. Materials and Methods

The research was divided into four phases. The methods, processes and materials used in each of the phases are presented below. The first two phases were based on previous research by some of the authors of this article, so this section focuses on the methods used in the last two phases, related to the experiments performed to evaluate LoRa in a transit vehicle tracking service setting.

### 3.1. Selection of a Suitable ITS Architecture for the Tracking Service

Selection of an ITS architecture for the transit vehicle tracking service is based on work presented in [13,15]. First, the ITS architecture proposed for the service [13], based on the American ITS architecture known as ARC-IT [34], was reviewed, which was. The proposed architecture for the PT01 service (PT01 is transit vehicle tracking in ARC-IT) was also reviewed. Additionally, the review included proposals for other reference architectures at an international level (such as European Intelligent Transport Systems Framework Architecture, often now known as The FRAME Architecture [35]) regarding the transit vehicle tracking service.

The use of a reference ITS architecture and the analysis performed to propose a specific ITS architecture for the service, which considered the special characteristics of the service (transit vehicle tracking service) in medium-sized cities in developing countries such as Colombia, is part of the novelty of the research presented in this paper.

The analysis of the aforementioned sources allowed to determine the service architecture, as shown in Figure 1. The proposed architecture includes three modules: transit vehicle OBE (on board equipment), ITS roadway equipment, and transit management center. It also proposes a system actor called system operator, who is in charge of interacting with the transit management center module, to obtain information on the state of the transit system. The transit vehicle OBE module is responsible for capturing information for each vehicle (location, speed, etc.) and transmitting it to the transit management center module through ITS roadway equipment, which features network devices (e.g., gateways) located near the roads, or at a medium distance. The architecture also features the basic layers proposed by IoT, where the device layer includes the transit vehicle OBE, the communications layer (between the transit vehicle OBE and the transit management center) is composed of two sections with the participation of the ITS roadway equipment module, and the processing layer is in the transit management center module.

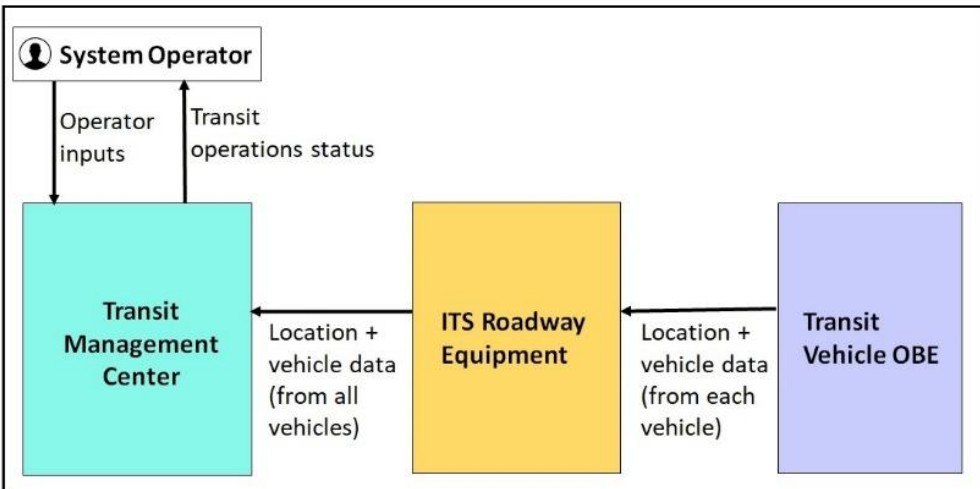

**Figure 1.** Selected intelligent transport system (ITS) architecture for transit vehicle tracking service.

## 3.2. Service Prototype Design

In [13], a proof of concept of the service was developed; this work presents a prototype, which is an incremental contribution to the development of the system. Considering the selected ITS architecture, a prototype of the system was built using LoRa between the transit vehicle OBE and ITS roadway equipment modules. Communication through LoRa is adequate between the mentioned modules, because several devices (one device per vehicle) transmit data packets (approximately 4–10 packets per minute) to a gateway. It was estimated that in a medium-sized city each gateway can receive packets of approximately 12–30 vehicles at a time, which can be easily handled by LoRa. Communication between the ITS roadway equipment module and the transit management center is done through the Internet, because the volume of data sent is greater and the distances may be longer, depending on the location of the transit management center (which is commonly located on the cloud). Figure 2 shows the prototype. The devices used in the transit vehicle OBE were an ESP32 LoRa Heltec card and an Ublox 6M GPS module. The selected gateway was the Dragino LG02, which allows the reception of LoRa messages. The recent versions of the Dragino gateways are mainly focused on the use of the LoRaWAN network protocol.

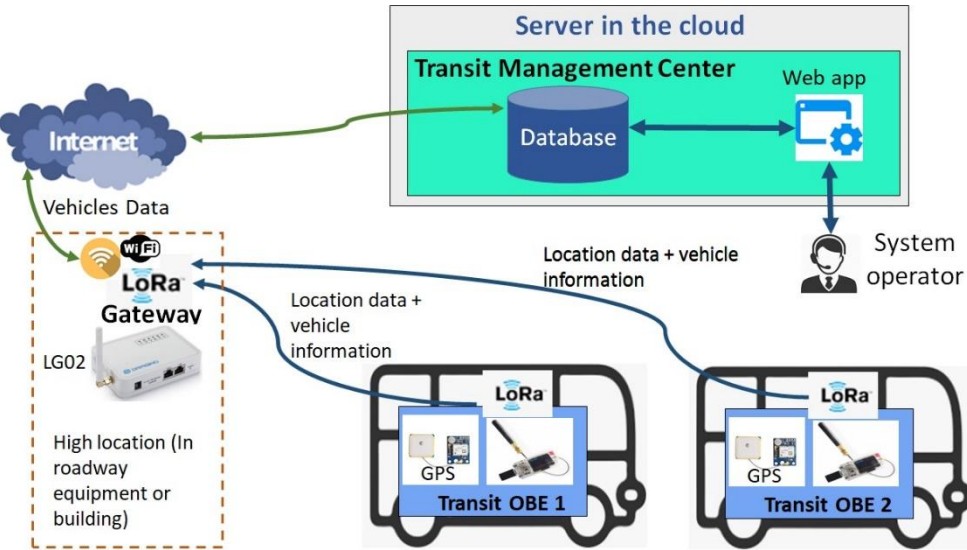

**Figure 2.** Transit vehicle tracking service prototype.

## 3.3. Design and Development of the LoRa Evaluation Experiments

Figure 2 presents the prototype setting for the performed experiments. The "web app" component was not implemented.

The characteristics of the performed experiments are different from those in the related works. This research used 72 settings in field tests, taking into account all the possible value combinations of the considered parameters (SF, BW, and CR). Although a few related works feature a similar or larger number of settings, the characteristics of the experiments (in terms of number of devices, packet rate, distance between sender and receiver, displacement of devices, and use of simulators) vary considerably. In addition, related works specifically related to tracking services in transportation have great limitations in terms of the used number of settings and the number of performed tests.

Six experiments were performed for evaluation of LoRa in the specified service. Three of them were performed with fixed distance, and three with variable distance. The characteristics of the six experiments are presented in Table 1.

**Table 1.** Characteristics of the performed experiments.

| Exp. Number | Exp. Type | Distance between GW and Devices (m) | Number of Devices | Number of Settings Used | Settings Not Used | Max. Speed (km/h) | Time between Packets (sec) |
|---|---|---|---|---|---|---|---|
| 1 | Fixed distance | 7 | 1 | 72 | 0 | 0 | 5 |
| 2 | Fixed distance | 20 | 1 | 72 | 0 | 0 | 5 |
| 3 | Fixed distance | 550 | 4 | 44 | 28 settings were discarded | 0 | 10 |
| 4 | Variable distance | 255–1245 (the vehicle traveled 1100 m) | 1 | 44 | 28 settings were discarded | 20 | 10 |
| 5 | Variable distance | 255–1245 (the vehicle traveled 1100 m) | 3 | 17 | 27 settings were discarded | 20 | 10 |
| 6 | Variable distance | 600–900 (the vehicle traveled 1800 m) | 3 | 17 | 27 settings were discarded | 60 | 7 |

The first three experiments were performed with fixed distances for two reasons. The first reason was to show a possible linear relationship between the measured RSSI value and the change in distance in any of the settings, or between the RSSI value and some of the parameters of the setting (SF, BW, and CR). This type of linear relationship would help to identify the appropriate settings for the service context. The second reason was to limit the number of settings for the moving vehicle experiments. With the experiments at fixed distances it was possible to rule out some of the 72 possible settings for the final experiments with variable distance. Testing for all settings would have required too much experimentation time, using settings that would not have yielded adequate results.

The first five experiments were performed in the city of Palmira (medium-sized city in the south-west of Colombia, South America). The gateway was located at the (3.548523 N, 76.288618 W) coordinates in the first five experiments. The last experiment (Experiment 6) was performed in the city of Popayán (another medium-sized city in the south-west of Colombia). The gateway in Experiment 6 was located at the (2.493120 N, 76.559722 W) coordinates. The change of location of the last experiment looked out for the best conditions in terms of needed materials, road conditions and the possibility of a better line of sight (LoS) between the gateway and the vehicles.

The details of the two types of experiments are discussed below.

### 3.3.1. Fixed Distance Experiments

Initially, three experiments were performed to verify the performance of LoRa subject to specific values of SF, BW, and CR, with the devices in fixed locations. These three initial experiments were designed to select the sets of LoRa settings with the best performance, to use them in the prototype setting (devices inside moving vehicles).

The initial experiments featured three different distances between the LoRa device and the gateway: 7 m, 20 m and 550 m. Figure 3 illustrates this. In the 550 m experiment, it was necessary to place the gateway at a height of 18.5 m in a building near a road.

In the first two experiments, all possible settings of the LoRa link were tested: SF was varied between 7 and 12, BW set to either 125, 250 or 500 kHz, and CR set to either 4/5, 4/6, 4/7 or 4/8, for a total of 72 possible combinations. For each setting, a total of 20 packets were sent from a LoRa device, with a pause of 5 s between packets. The following data was recorded for each one of the sent packets:

- Setting number
- Distance
- Spreading factor
- Bandwidth
- Coding rate
- RSSI
- Information in the received packet

A total of 44 settings out of the initial 72 were chosen to perform Experiment 3. Such chosen settings were the ones that had the best results in the two previous experiments, regarding the percentage of packets received and the average RSSI level received at the gateway. In Experiment 3, four LoRa devices were used. At the start of the experiment, packets were sent from only one of the LoRa devices and the other devices were added to the experiment one by one, every five seconds. The packet sending frequency was one packet every ten seconds. Twelve packets were sent from each one of the four LoRa devices. At the end, the devices were disconnected one by one, every five seconds. Once the delivery of the packets of the three devices for a setting was finished, a new setting was configured into each of the devices and the gateway, and packet delivery was started again. Figure 4 shows the location of the gateway and LoRa devices (on a map) in Experiment 3.

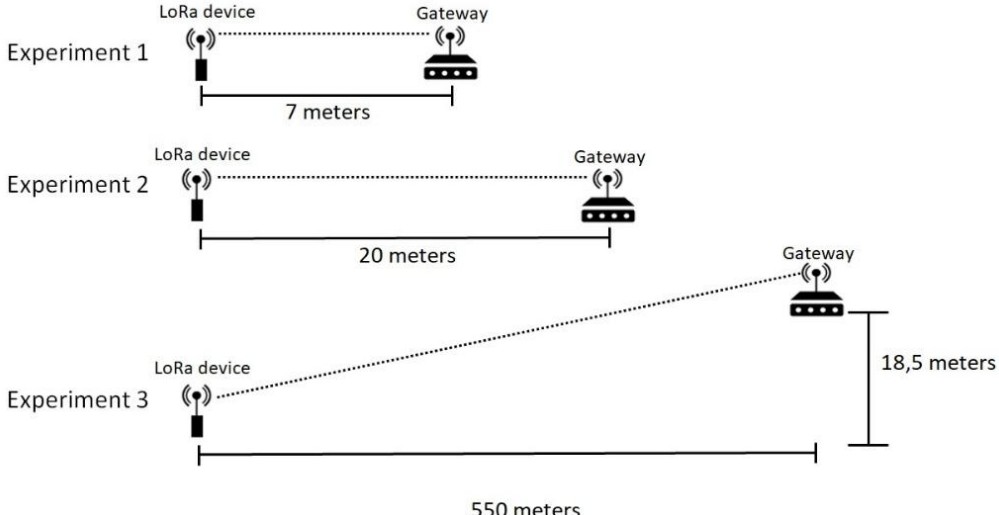

**Figure 3.** Distances and device locations in the three initial experiments (with fixed distances).

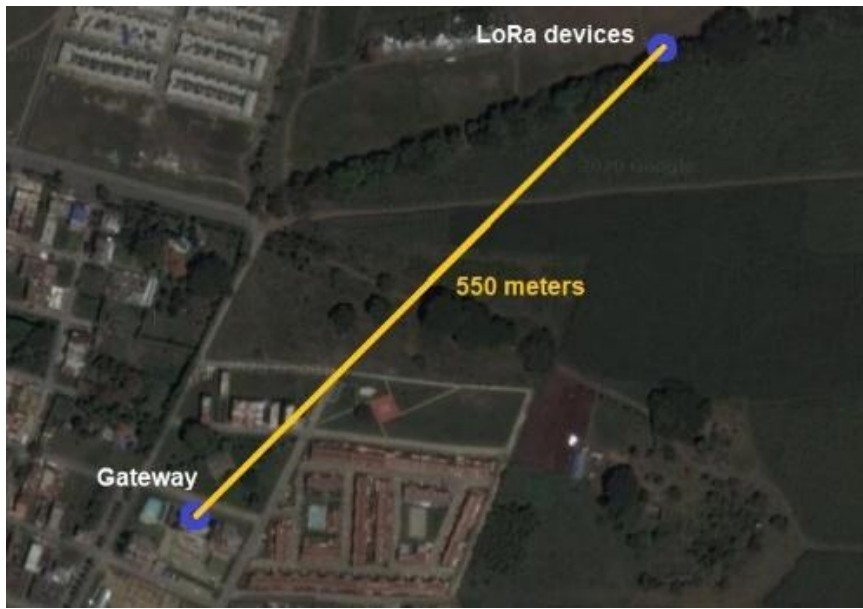

**Figure 4.** Location of the gateway and long-range (LoRa) devices from Experiment 3.

3.3.2. Variable Distance Experiments

In Experiment 4, only one LoRa device was used, and the distance between the gateway and the device ranged between 255 and 1245 m. The device was located in a vehicle, and 1100 m were driven

on a road at an approximate speed of 20 km/h (not exceeding 25 km/h to avoid issues due to Doppler effect [25]).

This experiment used the same 44 LoRa settings from Experiment 3. The route followed by the vehicle in Experiment 4 is presented in Figure 5 (red line). The figure also shows the distances between the gateway and the starting and ending points of the route (yellow lines).

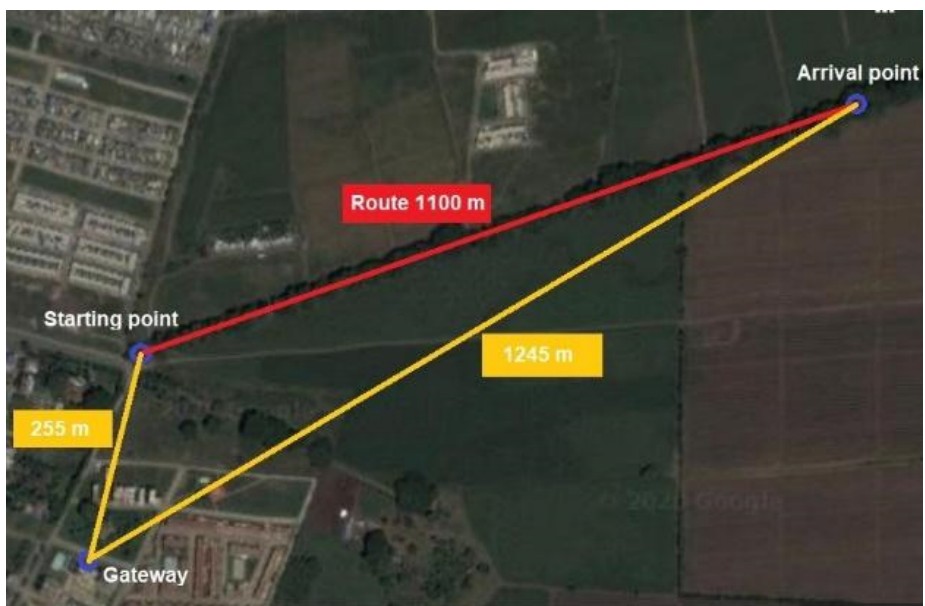

**Figure 5.** Vehicle route in Experiment 4 and distances to the gateway.

For each run of this experiment, the vehicle was located at the starting point and began its route to the arrival point at a maximum speed of 20 km/h; the drive took approximately 3 min and 30 s. One packet was sent every 10 s. Once the vehicle finished the route, packet sending was stopped and new parameters were configured for the next run.

Experiment 5 used the same route as Experiment 4. Three vehicles were used, and only 17 settings (the ones yielding the best performance in Experiment 4 in terms of successfully received packets and range) were used. One packet was sent every 10 s. Each vehicle started from the starting point every 15 s, so all three devices were transmitting 30 s into the experiment run. All vehicles were driven at a maximum speed of 20 km/h. Once all vehicles finished the route, packet sending was stopped, and the devices and vehicles were configured for the next run.

In the experiments it was considered important to have an environment as controlled as possible, so that the results obtained are affected as little as possible by external factors. For this reason, vehicle speed was limited to 20 km/h, to minimize Doppler effect impact. In addition, worth considering is the context of the service, since the transit service in intermediate cities of developing countries shares the road with other types of vehicles, which restricts vehicle speed to a range of 10 to 30 km/h for the most part of the route. In a few sections of the route, and depending upon traffic conditions, vehicles may reach 60 km/h, the city speed limit in Colombia.

The packet rate for Experiments 4 and 5 was determined considering the average and maximum vehicle speed. The distance traveled by a vehicle at the average speed (25 km/h) in 10 s is about 70 m, and the distance traveled at the speed limit (60 km/h) in 10 s is about 166 m. A maximum vehicle location error between 70 and 166 m was considered adequate, because the accuracy in this type of service is not critical for the user. In selecting the appropriate packet rate, it was determined that packets should be sent every three or more seconds, since the microcontroller card (ESP32 LoRa Heltec) requires this much time to read the data from the Global Positioning System (GPS)module and prepare

the data packet. In addition, for settings with a high SF the propagation time is higher, and packet collisions may occur at the gateway if the packet rate is too high.

Experiment 6 was designed to verify the impact on communications that may occur when vehicles travel at speeds between 20 and 60 km/h. The route followed by the vehicle in Experiment 6 is presented in Figure 6 (red arrows). The figure also shows the distances between the gateway and the starting and ending points of the route (yellow lines). Three vehicles were used, and 17 settings (the ones used in Experiment 5) were used. One packet was sent every 7 s (the frequency of sending packets was increased, considering that the speed of the vehicle was higher). Vehicles departed from the starting point every 15 s. All vehicles were driven at a maximum speed of 60 km/h. Once all vehicles finished the route, packet sending was stopped, and the devices and vehicles were configured for the next run.

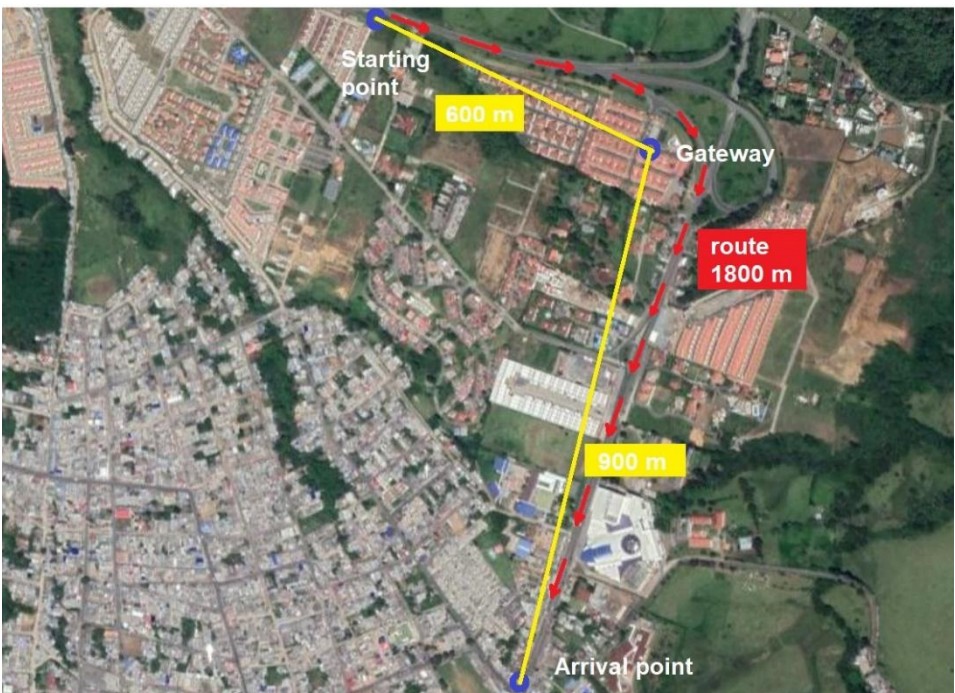

**Figure 6.** Vehicle route in Experiment 6 and distances to the gateway.

### 3.4. Data Processing and Result Evaluation

All the information collected in the six experiments was stored in spreadsheets and pre-processed to verify consistency. In some cases, packets with incorrect information (unexpected characters) were received, so a column indicating whether the packet was received correctly or not (packet state) was added to the spreadsheets.

Graphs of the collected information were made to identify the best settings in each experiment. Some contour graphs of the RSSI level were also made to identify the effect of the parameter values (BW, SF and CR). In some of the experiments, statistical analyses were performed to evaluate whether there was any linear relationship between some of the parameters and the RSSI value.

Regarding the last three experiments, results related to distance and number of LoRa devices were analyzed.

The analysis of experimental results featuring comparative graphs and tables to facilitate reader comprehension is considered part of the technical novelty of this work.

## 4. Results

This section presents the results obtained in the six experiments. The section ends with a comparison of the experimental results.

Considering, it is not possible to compare experiments performed in the related works with experiments in this research, because they have different characteristics, the SF = 9, BW = 125 kHz, CR = 4/5 setting was taken as a baseline, because it produced good results in some related works [17,27]. This baseline was used to evaluate RSSI and percentage of packets received of the best identified settings.

### 4.1. Experiment 1 Results: One Device, Distance 7 m

In this experiment, a large number of settings worked properly. For all settings, approximately 100% of the packets were received. However, in settings with high SF values (11 and 12), the packet payload was damaged. In those settings with a bandwidth of 125 kHz and an SF value of 10, the received data were also faulty, which suggests that at a short distance, with a high SF value, a BW greater than 125 kHz is needed to properly receive the information. Furthermore, with a low CR (4/5) packets were received at the gateway only with the lowest SF value (SF = 7); with values of 8, 9, 10, 11 and 12, all the received packets were damaged.

Figure 7 shows the results obtained in the ten best settings of Experiment 1, where 100% of the transmitted packets were received correctly.

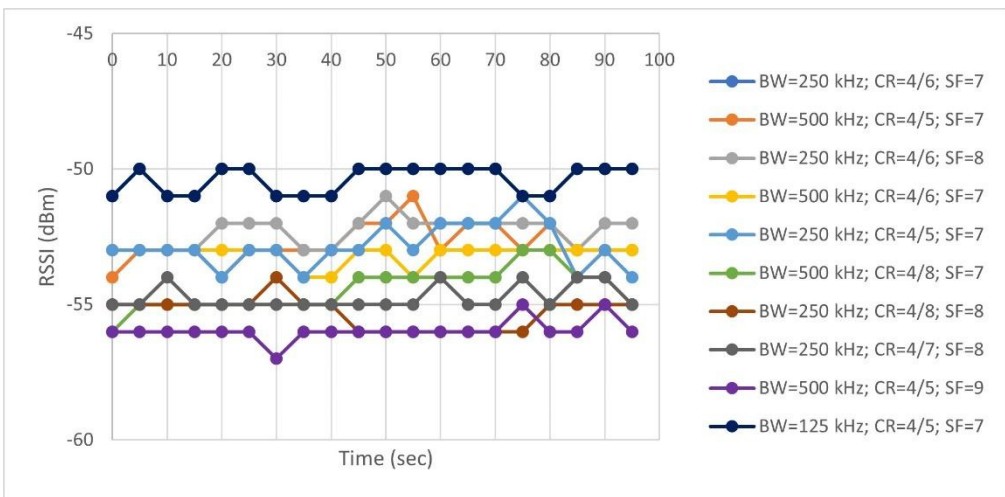

**Figure 7.** Receive signal strength indicator (RSSI) values for the best 10 settings in Experiment 1.

Figure 8 shows a contour graph for Experiment 1. The graph shows the effect of the different evaluated parameters on the RSSI, which is ideal when closer to zero. In the graph, values closer to zero are plotted in dark blue.

In the relationship between CR and BW, the most desirable values correspond to the lowest denominator for the CR, that is, 4/5. The BW value that is most suitable would be 250 kHz. This graph only features the denominator values of the CR in the *y*-axis for simplicity.

In the graph that relates the SF and the BW, the most desirable value for SF is 7, while for BW it is close to 500 kHz.

Finally, in the graph of the SF and CR parameters, the ideal values would be a SF of either 7 or 10, and a CR of 4/5 or 4/6. The *x*-axis shows only the denominator values for CR.

The baseline setting obtained an average RSSI value of −55.53 dBm and 100% of received packets, which was close to the best setting (SF = 7, BW = 250 kHz, CR = 4/6), with an average RSSI value of −50.36 dBm and 100% packets received.

Additionally, in this first experiment, an attempt was made to adjust the behavior of the RSSI level to a linear model combining the LoRa transmission parameters; however, the results were not satisfactory.

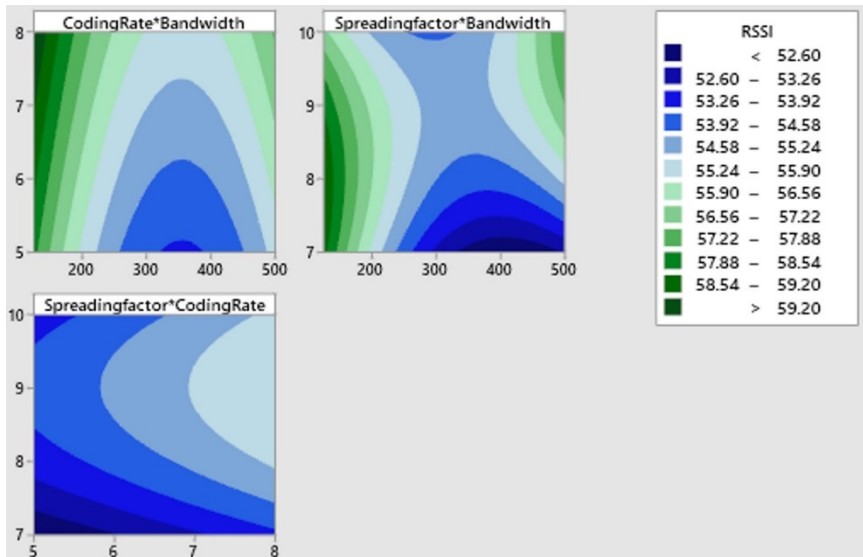

**Figure 8.** Contour graphs for settings evaluated in Experiment 1.

*4.2. Experiment 2 Results: One Device, Distance 20 m*

The results obtained in the experiment at 20 m (between gateway and LoRa device) showed that the best RSSI levels were obtained for high bandwidth values. The best RSSI levels were obtained on average in settings with a BW of 500 kHz.

In contrast with Experiment 1, packets were received at the gateway when BW = 125 kHz, with SF equal to 10 or 12, but in most cases the packets were damaged.

The percentage of packets received in each setting, as in the previous experiment, was approximately 100%. The cases where the data payload was corrupted must be considered, though.

Figure 9 shows the RSSI level for the best settings of this experiment. These values have a greater variability than those of Experiment 1, between −63 and −72 dBm.

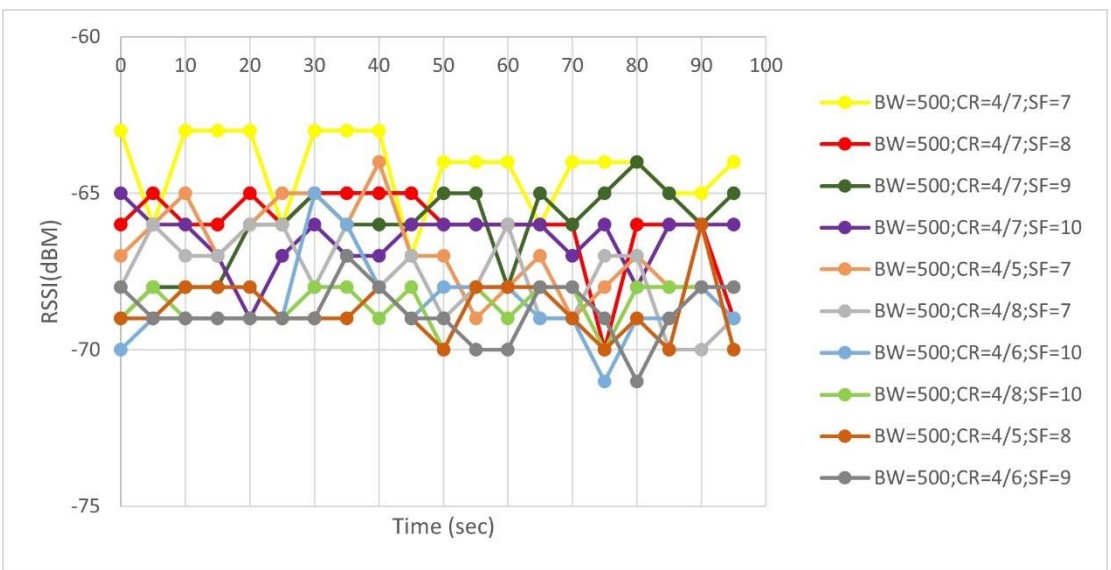

**Figure 9.** RSSI values for the best 10 settings in Experiment 2.

The baseline setting obtained an average RSSI value of −75 dBm and 100% of received packets, which was close to the best setting (SF = 7, BW = 500 kHz, CR = 4/7), with an average RSSI value of −64.20 dBm and 100% packets received.

In this second experiment, there was also an attempt to fit the behavior of the RSSI level to a linear regression model combining the LoRa transmission parameters. Results show that the relationship between RSSI and BW is statistically significant, with a confidence level of 90% and shows a stable behavior. The linear equation found was: RSSI = 78.595 − 0.02155 BW.

*4.3. Experiment 3 Results: Four Devices, Distance 550 m*

The results obtained in Experiment 3, featuring a distance of 550 m between the gateway and four LoRa devices, showed that the only setting where 100% of the packets were received was BW = 125 kHz, CR = 4/6 and SF = 7. Analyzing this setting in the previous experiments, Experiment 1 featured some damaged packets and a high RSSI average (−58 dBm), compared to the best setting in Experiment 2, with an average RSSI value of −50 dBm. In Experiment 2, this setting had an acceptable behavior, meaning that all the transmitted packets were received correctly, but it did not stand out as a setting with the best average RSSI values (the value for this setting was −73 dBm, when the highest average value of Experiment 2 was −66 dBm approximately).

It is important to note that this experiment considered only 44 settings, by discarding 28 settings that did not yield good results in Experiments 1 and 2. The discarded settings include those with SF equal to 11 and 12 (24 settings) and the settings with an SF of 10 and a BW of 125 kHz (4 settings).

There is a considerable decrease in the percentage of received packets in most of the settings. This is explained because more than one LoRa device is being used at the same time (most of the time, four in each setting) and the four devices have the same parameter values (mainly SF), which can generate packet collisions, according to [19]. Only in 25% of the settings 100% of the packets were received; while 47% of the settings featured a reception rate between 99% and 75%, and 23% of the settings featured a reception rate below 75%.

The behavior of the best RSSI level when increasing the BW is stable.

The baseline setting obtained an average RSSI value of −105.75 dBm and 91.67% of received packets, which was close to the best setting (SF = 7, BW = 125 kHz, CR = 4/6), with an average RSSI value of −103.27 dBm and 100% packets received.

The settings with the better RSSI levels and the better percentages of packets received in Experiment 3 are presented in Table 2.

**Table 2.** Better settings in Experiment 3.

| Bandwidth (kHz) | Coding Rate | Spreading Factor |
|:---:|:---:|:---:|
| 125 | 4/6 | 7 |
| 125 | 4/8 | 7 |
| 500 | 4/5 | 10 |
| 250 | 4/6 | 9 |

Figure 10 shows the behavior of the best setting of this experiment, which shows some variability in the RSSI levels obtained for each of the four devices (approximately 8 dBm).

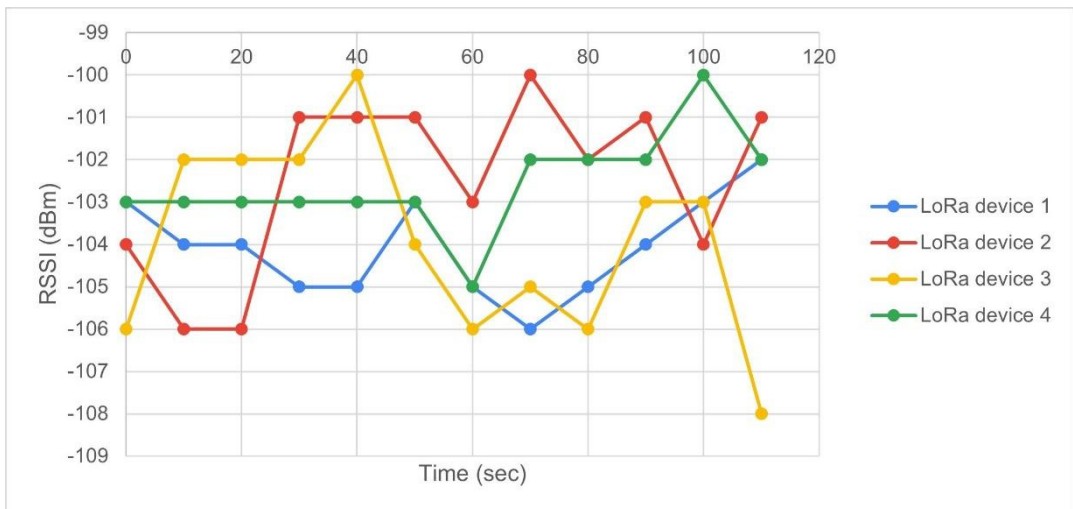

**Figure 10.** RSSI values for the best setting (spreading factor (SF) = 7, BW = 125 kHz, coding rate (CR) = 4/6) in Experiment 3.

Figure 11 presents the contour graphs for Experiment 3. As in Figure 8, only the denominator values of the CR are presented.

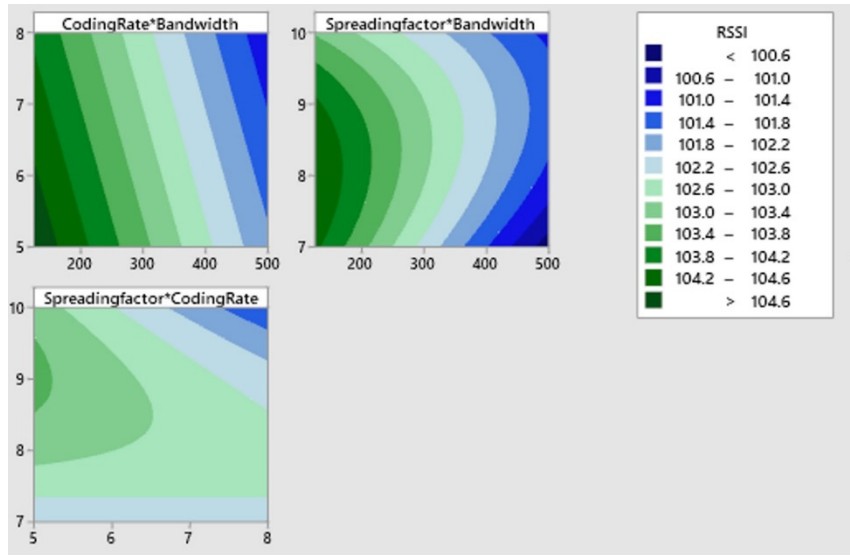

**Figure 11.** Contour graphs for settings evaluated in Experiment 3.

The first graph (relationship between CR and BW) shows that the most desirable values of RSSI are obtained with CR settings close to 4/8, and a BW close to at 500 kHz. The second plot (relationship between SF and BW) shows that a low SF (7) and a BW close to 500 yields the most desirable RSSI. Finally, the third graph (relationship between SF and CR) shows that there is no specific area for ideal RSSI values, when compared to the other graphs. However, CR = 4/8 and SF = 10 yields the best RSSI levels.

### 4.4. Experiment 4 Results: One Device, Distance between 255–1245 m

In this experiment, the same 44 settings from Experiment 3 were used. In most of the settings, packets at distances greater than 650 m were not received at the gateway, mainly because of obstacles in the line of sight (LoS). Figure 12 shows RSSI levels with respect to distance for some of the best settings, which were obtained with a low BW (125 kHz), a CR with low denominator (4/5) and a SF of 7, 8 or 9.

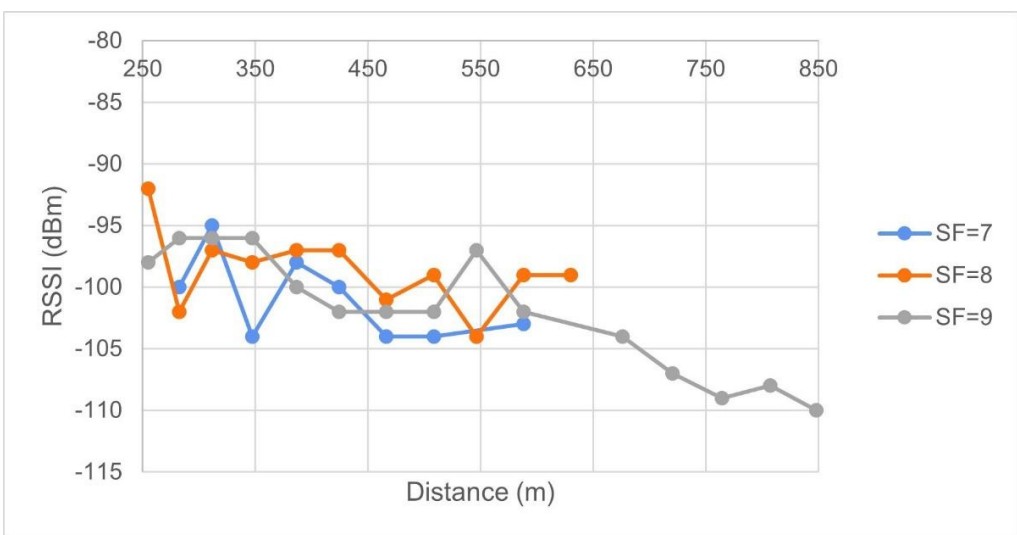

**Figure 12.** RSSI versus distance with bandwidth (BW) = 125 kHz and CR = 4/5, in Experiment 4.

Although there is some variability in the RSSI levels obtained for the three settings (see Figure 12), the range of values in which most of the values vary (before 650 m) is less than 15 dBm. In settings with an SF of 10 packets were received above 750 m, however the RSSI decreased by a few dBm. It was possible to validate in some other cases that with a high SF, higher distances can be achieved with a lower RSSI. Figure 13, shows that with SF = 9 (BW = 125 kHz and CR = 4/6) a greater distance was reached.

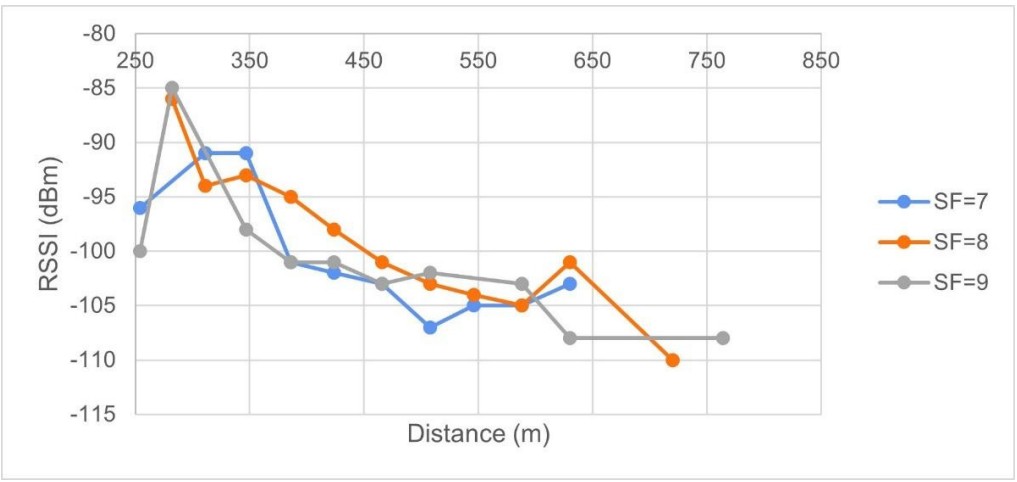

**Figure 13.** RSSI versus distance with BW = 125 kHz and CR = 4/6, in Experiment 4.

There were some other settings (with BW = 125 kHz), where the maximum distance was not reached with a high SF. Figure 14 shows that maximum distance for BW = 125 kHz was reached with a SF = 7.

The variation of SF in the maximum distance reached was also observed with the values of BW equal to 250 kHz and 500 kHz, where in some cases the maximum distance was reached with SF = 9 or SF = 10, but in other cases it was reached with SF = 7.

With respect to percentage of received packets, no settings yielded 100% reception. A total of 27% of the settings featured a reception rate between 99% and 75%; 10% of the settings featured a reception rate below 50%.

The baseline setting was the best for this experiment. It obtained an average RSSI value of −103 dBm, 94% packets received, and a maximum distance of 848 m.

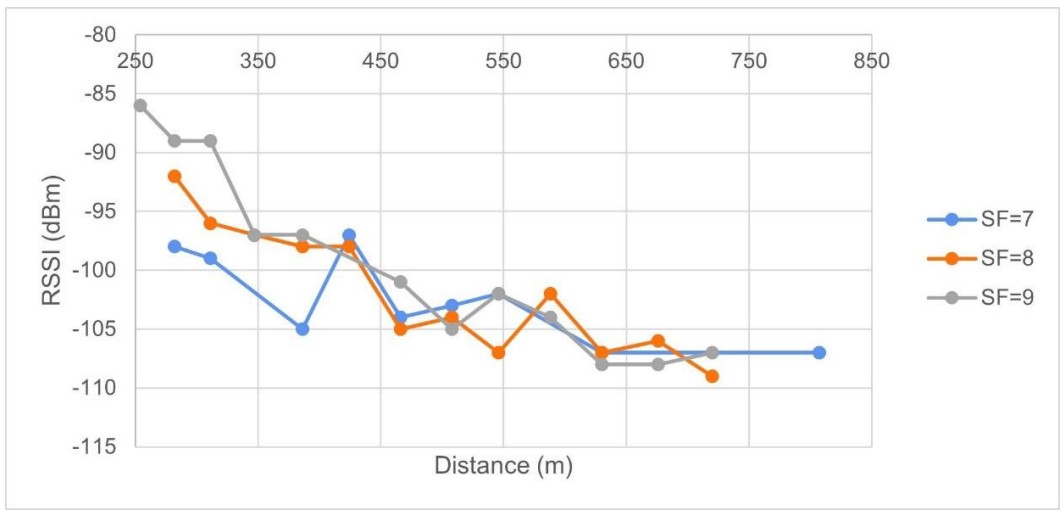

**Figure 14.** RSSI versus distance with BW = 125 kHz and CR = 4/7, in Experiment 4.

*4.5. Experiment 5 Results: Three Devices, Distance between 255–1245 m*

In this experiment, only 17 settings were used, discarding some of the settings that did not have good results in Experiment 4.

Some settings such as SF = 9, BW = 250 kHz, CR = 4/5 (S1), and SF = 9, BW = 250 kHz, CR = 4/8 (S2) performed well both in Experiment 4 and in Experiment 5. The first setting was one of the three best settings in each of the experiments, while the second setting was not within that group. The results obtained in these settings in both experiments (4 and 5) are presented in Table 3.

**Table 3.** Better settings in both Experiments 4 and 5.

| Measured Parameters | S1 (SF = 9, BW = 250 kHz, CR = 4/5) | | S2 (SF = 9, BW = 250 kHz, CR = 4/8) | |
|---|---|---|---|---|
| | Experiment 4 | Experiment 5 | Experiment 4 | Experiment 5 |
| Packets received at the gateway (%) | 87 | 81 | 81 | 73 |
| Max. distance for packet reception (meters) | 848 | 848 | 676 | 807 |
| RSSI level (dBm) | −94 | −100 | −96 | −99 |

There were also some settings that had a very good percentage of received packets in Experiment 4 and showed a considerable reduction of this indicator in Experiment 5. For example, the setting SF = 9, BW = 125 kHz, CR = 4/5 (going from 94% in Experiment 4 to 65% in Experiment 5) and setting SF = 8, BW = 125 kHz, CR = 4/8, (going from 94% in Experiment 4 to 60% in Experiment 5).

Although, there were cases of inverse behavior (better in Experiment 5 than in Experiment 4) the differences were not too significant. For example, the setting with SF = 7, BW = 500 kHz, CR = 4/8 of Experiment 4 presented bad indicators (percentage of packets received 44%, maximum distance 424 m and average RSSI −94 dBm), while in Experiment 5, two of these indicators improved a little (percentage of packets received 58.3%, maximum distance 630 m and average RSSI −98 dBm).

For identifying the possible causes of these results, and also discarding the possibility that these results have been obtained with the influence of some type of event or arbitrary conditions, this experiment was repeated under the same conditions (configured parameters, number of devices, and distances).

The baseline setting obtained an average RSSI value of −103 dBm, 64.6% of received packets, and 720 m of maximum distance. These results were close to the best setting, except for the percentage of packets received (SF = 9, BW = 250 kHz, CR = 4/5), with an average RSSI value of −100 dBm, 81% packets received, and a maximum distance of 848 m.

Regarding the RSSI, Figure 15 shows an example for one of the settings with the best percentage of packets received and distance reached.

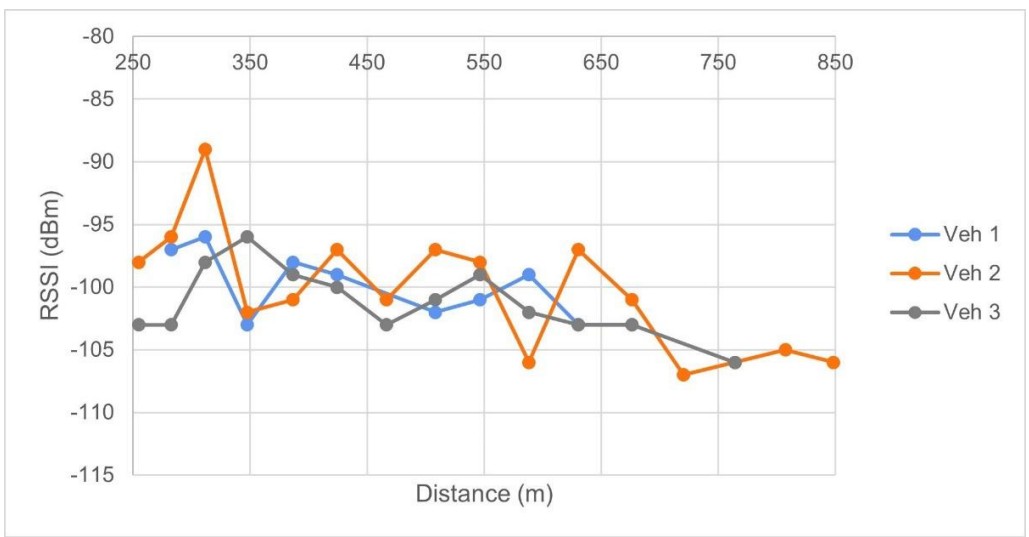

**Figure 15.** RSSI values obtained for Experiment 5, in the setting SF = 9, BW = 250 kHz and CR = 4/5.

### 4.6. Experiment 6 Results: Three Devices, Distance between 600–900 m

In this experiment, only 17 settings were used, using the same settings and number of devices as in Experiment 5. It is important to bear in mind that the location of the gateway and the route were modified for this experiment, looking for the best conditions. The route of this experiment was longer (1800 m), the route included a curve and the height of the gateway location was decreased to 6 m.

The main objective of this experiment was to determine the influence of vehicle speed on the communication operation using LoRa. The maximum speed used in Experiments 4 and 5 was 20 km/h, while for Experiment 6 the vehicles traveled at a maximum speed of 60 km/h. The vehicles traveled at maximum speed for approximately 600 m, however the average speed was 40 km/h.

The results obtained in this experiment were similar to those obtained in Experiment 5, only a few settings had a better RSSI level (SF = 9, BW = 500, CR = 4/6 and SF = 10, BW = 250, CR = 4/6); the maximum distance of this experiment (890 m) was achieved with the setting SF = 10, BW = 250, CR = 4/6; and the maximum percentage of packets received (75%) was achieved with the setting SF = 10, BW = 250, CR = 4/7. The most affected parameter in this experiment was the percentage of packets received, which decreased on average by approximately 6%. This average decrease in the percentage of packets received can be accounted to factors such as the Doppler effect.

The baseline setting obtained an average RSSI value of −105 dBm, 67.35% of received packets, and 810 m of maximum distance. The results were close to the best setting (SF = 10, BW = 250 kHz, CR = 4/6), with an average RSSI value of −100 dBm, 72% packets received, and a maximum distance of 890 m.

### 4.7. Comparison of the Results Obtained for the Six Experiments

The comparative analysis began by observing the behavior of the best and the worst setting in Experiment 1 (distance 7 m) compared to Experiment 2 (distance 20 m), that showed how the results varied when changing the distance a little. Similarly, the comparison was made between Experiment 2 and Experiment 3 (distance 550 m). Next, the results obtained from the best and worst settings of Experiment 3 were compared with their performance in the 7 m and 20 m distance experiments. Lastly, the results in the variable distance experiments were evaluated in the three best settings of the three initial experiments. In addition, the best settings were evaluated in Experiments 4, 5 and 6 with respect to the percentage of packets received and the greatest distance reached by a packet.

In Experiment 1, the setting corresponding to SF = 7, BW = 250 kHz, CR = 4/6 yielded the best average RSSI value (−50.36 dBm). This setting, at 20 m, had an average RSSI value of −73.40 dBm, located approximately in the middle of the settings ordered under the same criteria. Therefore,

by increasing the distance, this setting worsened its signal level received at the gateway much more than expected. The worst setting of Experiment 1 was SF = 7, BW = 125 kHz, CR = 4/8, its average RSSI value was −62.66 dBm, a performance that was maintained when the distance was scaled to 20 m, placing it between the ten worst settings.

In Experiment 2, the best setting was SF = 7, BW = 500 kHz, CR = 4/7, with an average RSSI value of −64.20 dBm; this same setting in Experiment 3 was one of the 5 worst settings. Furthermore, the worst setting of Experiment 2, which corresponded to the parameters SF = 9, BW = 125 kHz, CR = 4/8, maintained its low average RSSI level in Experiment 3.

In Experiment 3 the only setting where 100% of packets were received was SF = 7, BW = 125 kHz, CR = 4/6. For this same setting in Experiment 1, all packets were also received, despite the fact that some were corrupted, and it maintained a high average RSSI, −57.71 dBm. This same setting in Experiment 2 had an acceptable behavior, meaning that all the transmitted packets were received in good condition, although the average RSSI level was not very high.

According to what has been evaluated so far, the best settings for the first three experiments were SF = 7, BW = 250 kHz, CR = 4/6; SF = 7, BW = 500 kHz, CR = 4/7; and SF = 7, BW = 125 kHz, CR = 4/6. These settings in Experiment 4 showed 62%, 50% and 62% of packets received respectively, being settings with low results, compared to others in this experiment. For that reason, these settings were among those that were not evaluated in Experiments 5 and 6.

Table 4 presents the three best settings regarding RSSI signal level of the first three experiments. In addition, the three settings with the best percentages of packets received in Experiment 3 are presented.

**Table 4.** Best settings in Experiments 1, 2, and 3.

| | Best Settings Regarding RSSI in Exp. 1 | | | Best Settings Regarding RSSI in Exp. 2 | | | Best Settings Regarding RSSI in Exp. 3 | | | Best Settings Regarding % of Received Packets in Exp. 3 | | |
|---|---|---|---|---|---|---|---|---|---|---|---|---|
| SF | 7 | 7 | 7 | 7 | 8 | 9 | 7 | 8 | 9 | 7 | 9 | 7 |
| BW | 250 | 500 | 500 | 500 | 500 | 500 | 500 | 500 | 500 | 125 | 250 | 125 |
| CR | 4/6 | 4/7 | 4/5 | 4/7 | 4/7 | 4/7 | 4/7 | 4/8 | 4/6 | 4/6 | 4/6 | 4/8 |

Taking into account Table 4, it can be stated that in experiments with fixed distances it is convenient to use a BW of 500 kHz, an SF of 7, 8 or 9 and a CR of 4/5, 4/6 or 4/7 to obtain an adequate level of RSSI. However, when considering the percentage of packets received in a setting with more than one node, a lower bandwidth is convenient and SF should be 7.

Table 5 shows the best settings for Experiments 4, 5, and 6 (independently), with respect to the percentage of packets received. Additionally, for these settings, the data obtained from the average RSSI and the maximum distance reached by a packet received at the gateway are shown.

According to the results presented in Table 5, the setting SF = 9, BW = 250 kHz, CR = 4/5 performed well in Experiments 4 and 5. This also held for Experiment 6, so this is an adequate option to perform a greater number of experiments in subsequent research, considering also that this setting presented good results in Experiments 1, 2 and 3, although in Experiment 3, two of the four nodes presented a low percentage of received packets for this setting. The difference in percentage of packets received between Experiments 4 and 5 (in the best settings), shows that by increasing the number of LoRa devices (from 1 to 3), the percentage of packets received decreases between 10% and 20%. The difference in percentage of packets received between Experiments 5 and 6 (in the best settings), shows that by increasing the maximum speed, the percentage of packets received decreases between 3% and 10%.

The baseline setting (SF = 9, BW = 125, CR = 4/5) had very similar results to those presented by the best setting in each of the six experiments. This determines that the baseline setting, although it is not the best setting in all the experiments, had a good overall performance in all of them.

**Table 5.** Best settings in Experiments 4, 5, and 6 (independently).

| Measured Parameters | Best Settings in Exp. 4 | | | Best Settings in Exp. 5 | | | Best Settings in Exp. 6 | | |
|---|---|---|---|---|---|---|---|---|---|
| SF | 9 | 8 | 9 | 9 | 9 | 8 | 10 | 9 | 9 |
| BW (kHz) | 125 | 125 | 250 | 250 | 250 | 250 | 250 | 250 | 250 |
| CR | 4/5 | 4/8 | 4/5 | 4/5 | 4/6 | 4/7 | 4/7 | 4/8 | 4/5 |
| Packets received (%) | 94 | 94 | 87 | 81 | 81 | 73 | 75 | 72 | 71 |
| Average RSSI (dBm) | −103 | −98 | −94 | −100 | −101 | −98 | −102 | −105 | −103 |
| Max. packet distance (meters) | 848 | 807 | 848 | 848 | 764 | 676 | 790 | 860 | 820 |

## 5. Discussion

The results obtained in the six experiments show the behavior of LoRa transmission (with respect to the percentage of packets received and the RSSI level) is quite variable when the distance, and the number of LoRa devices are scaled, using the same values for SF, BW and CR. It was not possible to determine a setting that had a constant behavior for all the tested distances and for different number of nodes. However, one setting had an adequate behavior in most of the experiments, mainly in the final three: SF = 9, BW = 250 kHz, CR = 4/5, where the percentage of packets received was between 87%, 81%, and 75% the RSSI level was acceptable (between −95 and −103 dBm) and the maximum distance reached was adequate (between 848 and 820 m). This setting is important, because it scaled well in terms of distance, number of vehicles and vehicle speed in most of the six performed experiments (except for Experiment 3).

Results obtained in each of the experiments are relevant, because they identify some suitable transmission settings of LoRa for each of the types of experiments, in addition to presenting the ranges of RSSI and percentage of packets received. This information can be useful for implementing services using this communication technology. The detailed information of the results obtained in each of the settings of each of the experiments can be consulted with the authors to collaborate in related researches.

The linear relationship identified between the RSSI level and the BW in Experiment 2 may not be very useful for a mobility service such as transit vehicle tracking service, however for other types of services in which LoRa is used at a distance of approximately 20 m between devices and gateway, this relationship may be relevant, being able to predict (with some accuracy, approximately 70%) an expected RSSI value for a given BW.

Although in the experiments with variable distance (Experiments 4, 5, and 6) the maximum distance reached by a packet received by the gateway was relatively low, this may be mainly due to the fact to the interruption of LoS. It was shown that direct LoS between devices has a strong influence in the operation of LoRa. In the process of searching for suitable locations for the gateway and the route of the vehicles, several options were tested. When there was no adequate LoS (too many obstacles), the LoRa gateway stopped receiving packets. The best distance reached in experiments 4−6 was 890 m. This can be improved by placing the gateway in a high enough place, which was not possible in the used test environments. However, with a linear distance to the gateway of 890 m, a vehicle could make a route of several kilometers around the gateway (with an adequate LoS), which is positive regarding the transmission range in the proposed service.

The decrease on the percentage of packets received when the number of nodes was increased from 1 to 3, or from 1 to 4, was approximately between 10% and 20%. To improve this percentage, future work could be done to configure the LoRa devices (operating at the same time) with different SFs, to minimize the probability of packet collisions. Another option that can improve the percentage of received packets is the dynamic and automatic variation of the parameters during transmission, using network protocols such as LoRaWAN or algorithms that allow such variation.

A total of 6% less packets were received in the increased speed scenario. This could be considered manageable, because transit vehicles in the evaluated context reach this speed range (20–60 Km) in very few sections of their route and during short periods of the day.

Regarding the baseline setting, it is highlighted that the RSSI, percentage of packets received, and maximum distance values are considerably close to the values obtained by the best settings in each experiment. This means that this baseline setting should also be considered in future works to perform a larger number of tests for the same or similar services.

A mobility service such as the one researched in this work requires the most stable behavior possible of the communication technology at short, medium and long distances, also with a low or high number of LoRa devices sending messages at the same time. The most convenient setting for this type of service was identified by good results in almost all experiments, mainly in experiments 4, 5, and 6 (SF 9, BW = 250 kHz, CR = 4/5). Considering that the maximum distance reached in this setting was relatively low (820–848 m), it is recommended to locate the gateways required for communication at points adjacent to the roads, and at a high enough place to have an adequate LoS over a much greater distance.

The results are considered relevant because a significant number of possible LoRa settings were evaluated, considering all the combinations of SF, BW, and CR. 72 settings were evaluated in the first two experiments, 44 settings in Experiments 3 and 4 and 17 settings in Experiments 5 and 6, discarding the settings that did not have adequate results in the previous experiments. In addition, parameters such as distance, number of devices and vehicle speed were scaled.

For future testing, it is recommended to use both the baseline setting and the setting with the best results in this work, to further increase the number of devices (eight or more), to use more than one gateway, and to use different routes for vehicles, to get even closer to a real operational environment. Another important aspect in the additional tests is to be able to use a mechanism that allows dynamic modification of the LoRa communication parameters, mainly SF and BW, to allow for changing conditions such as distance, presence of obstacles or number of devices. One possibility to do this is by using the LoRaWAN protocol, or some dynamic allocation algorithms.

## 6. Conclusions

A transit vehicle tracking service prototype was developed based on an ITS architecture that facilitates interoperability with other mobility services that consider the used architecture as a base. The prototype was built using LoRa between the critical modules. Six experiments were performed to evaluate the operation of LoRa in the prototype. The evaluated parameters in the experiments were spreading factor (SF), bandwidth (BW), and coding rate (CR), and the ideal values found for the service were SF = 9, BW = 250 kHz, CR = 4/5. This setting of parameters scaled well in terms of distance, number of vehicles, and vehicle speed in most of the six performed experiments. The results obtained were compared to a baseline (SF = 9, BW = 125, and CR = 4/5) obtaining similar values in the best setting of each experiment. Some aspects affecting the results were evaluated: (a) LoS between devices and gateway, which seriously affects operation; (b) number of devices; by increasing this number, the percentage of lost packets increased between 10% and 20%; and (c) the increase in vehicle speed (from 20 to 60 km/h) causes a lost packets increase of 6%. In future works, it is recommended to validate with additional tests the operation of LoRa for the tracking service, using the best identified setting. Additionally, it is recommended to work on a mechanism that allows dynamically modifying LoRa parameters.

**Author Contributions:** Conceptualization, R.S.-C. and Á.P.d.l.C.; methodology, F.J.M., J.S.Q.Y., A.D.V.L., R.S.-C., and Á.P.d.l.C.; validation, F.J.M., J.S.Q.Y., A.D.V.L., and R.S.-C.; formal analysis, F.J.M., J.S.Q.Y., A.D.V.L., and R.S.-C.; investigation, F.J.M., J.S.Q.Y., A.D.V.L., and R.S.-C.; resources, Á.P.d.l.C. and J.M.M.M.; writing—original draft preparation, R.S.-C., F.J.M., J.S.Q.Y., and A.D.V.L.; writing—review and editing, Á.P.d.l.C. and J.M.M.M.; supervision, Á.P.d.l.C. and J.M.M.M.; All authors have read and agreed to the published version of the manuscript.

**Funding:** This research received no external funding.



**Acknowledgments:** Authors wish to thank Universidad Icesi (ICT Department) and Universidad del Cauca (Telematics Department) for supporting this research.

**Conflicts of Interest:** The authors declare no conflict of interest.

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
