# Peer review of "Experimental Evaluation of LoRa in Transit Vehicle Tracking Service Based on Intelligent Transportation Systems and IoT"

_electronics, doi:10.3390/electronics9111950_

Round 1

Reviewer 1 Report

  • Why were fixed distance tests perfromed when you are evaluating a transport system
  • In experiment 2.3.2 you say "not exceeding 25 km/h to avoid issues due to Doppler effect". In many settings this is very slow and therefore not representative of driving conditions. 
  • You also say "One packet was sent every 10 seconds", is this enough to keep track of vehicle location in a real world setting? Why was this packet rate selected?
  •  

Reviewer 2 Report

The authors stated that the paper addresses c and d, that is "c) Designing and developing experiments to evaluate LoRa in the prototype; d) Obtaining the optimal LoRa parameters for the service."

However, in [13], the system has been implemented an tested as a proof of concept. This makes the contribution only in (b) which is incremental. 

Also, several parameters in the paper are not justified. E.g. why did you consider the speed of 20km/h? in practical systems, tracking should work for low, average, and high speed (ranging from 30km/h till 180 km/h). 

One final point is that the conclusion is badly written: It is too long and didn't follow the academic way.

Reviewer 3 Report

This work reports an experimental study about Low Power Wide Area Networks. Even though the background and introduction are explained adequately, this work lacks a detailed related works section. Furthermore, the technical novelty of the work is not adequate/not shown clearly.

Major issues:

  • lack of a related works section
  • lack of comparison to a baseline or an alternative method

Round 2

Reviewer 2 Report

Thanks for clarifying your manuscript. 

The conclusion has been improved however, in my opinion, it is still long. I advise that you shorten this to one paragraph and push the remaining parts to the discussions section.

Reviewer 3 Report

The revised version of the paper addresses the major concerns I had with the initial submission. There are only some minor issues left.

[Minor issues]

-One remaining minor issue would be table formatting. Especially, Table 1 has a lot of white space.

-Figures 9, 10, 13, 14, 15: How did these curves are fitted? As far as I understand, you have discrete observations and have not used a curve-fitting algorithm to obtain continous values. A scatter plot or a line graph can be a better choice here.  
